# Analysis of the Feeding Behavior and Life Table of *Nilaparvata lugens* and *Sogatella furcifera* (Hemiptera: Delphacidae) under Sublethal Concentrations of Imidacloprid and Sulfoxaflor

**DOI:** 10.3390/insects13121130

**Published:** 2022-12-07

**Authors:** Yeolgyu Kang, Hyun-Na Koo, Hyun-Kyung Kim, Gil-Hah Kim

**Affiliations:** Department of Plant Medicine, College of Agriculture, Life and Environment Science, Chungbuk National University, Cheongju 28644, Republic of Korea

**Keywords:** *Nilaparvata lugens*, *Sogatella furcifera*, imidacloprid, sulfoxaflor, electrical penetration graph, life table

## Abstract

**Simple Summary:**

The brown planthopper (BPH), *Nilaparvata lugens* and white-backed planthopper (WBPH) *Sogatella furcifera* are major rice pests in many parts of Asia. This study was conducted to elucidate the action mechanisms in sublethal exposures of imidacloprid and sulfoxaflor on the feeding behavior of the planthopper. The sublethal concentrations of imidacloprid and sulfoxaflor inhibited the phloem feeding of the BPH and WBPH and decreased their reproduction longevity. Additionally, sulfoxaflor was effective in inhibiting feeding in the BPH, and imidacloprid was more effective in inhibiting feeding in the WBPH. Therefore, sublethal effects of insecticide vary according to insect pest species.

**Abstract:**

The brown planthopper (BPH) *Nilaparvata lugens* and white-backed planthopper (WBPH) *Sogatella furcifera* are serious rice insect pests that cannot overwinter in Korea and migrate from southeast Asian countries and China. In this study, we investigated the sublethal effects of imidacloprid and sulfoxaflor on the biological parameters and feeding behavior of planthoppers. These sublethal concentrations significantly decreased the net reproduction rate (*R*_0_), the intrinsic rate of increase (*r_m_*), and the mean generation time (*T*). For BPHs, the total durations of nonpenetration (NP) waveform by imidacloprid (LC_10_ = 164.74 and LC_30_ = 176.48 min) and sulfoxaflor (LC_10_ = 235.57 and LC_30_ = 226.93 min) were significantly different from those in the control group (52.73 min). In addition, on WBPHs, the total durations of NP waveform by imidacloprid (LC_10_ = 203.69 and LC_30_ = 169.9 min) and sulfoxaflor (LC_10_ = 134.02 and LC_30_ = 252.14 min) were significantly different from those in the control group (45.18 min). Moreover, the LC_10_ and LC_30_ of these insecticides significantly decreased the phloem feeding time. In conclusion, imidacloprid had a better effect on the inhibition of feeding of the WBPH, and sulfoxaflor showed a better effect on the inhibition of feeding of the BPH. Therefore, it is suggested that treatment with sublethal concentrations of the above insecticides will reduce the feeding of BPHs and WBPHs on rice phloem.

## 1. Introduction

Rice is a major food crop in the world and is essential for global food security [1]. The brown planthopper (BPH) *Nilaparvata lugens* and white-backed planthopper (WBPH) *Sogatella furcifera*, which are insect pests that cause problems in rice production, occur in Asia and cannot overwinter in Korea. Therefore, they occur in temperate regions of east Asia, such as Korea and Japan, after moving long distances from early summer every year in countries such as China and Vietnam, which are overwintering regions [2,3,4,5]. Since wintering is impossible in Korea, the BPH and WBPH come to Korea via long-winged forms. They cause hopperburn through continuous feeding, turning the rice yellow and drying the rice quickly, and a wide range of rice turns brown [6]. Rice suffers from sucking damage by the BPH and WBPH, its normal growth is disturbed, and crop yields can be reduced, resulting in economic losses of billions of dollars per year [7,8,9]. WBPHs can transmit the virus diseases of rice known as the SRBSDV (Southern rice black-streaked dwarf virus), and BPHs transmit the RGSV (Rice grassy stunt virus) and RRSV (Rice ragged stunt virus) [10,11]. In addition, BPHs can cause the hopperburn which can reduce the 1000-grain weight [12].

One of the methods for controlling these pests is chemical control, and pesticides are mainly used. Pesticides such as imidacloprid, sulfoxaflor, clothianidin, and dinotefuran have been registered in the control of the BPH and WBPH [13]. Imidacloprid (4a) and sulfoxaflor (4c) act on nicotinic acetylcholine receptors to repeatedly excite insect nerve cells and cause the receptor to lose its function, leading to death of the insect [14,15]. The use of insecticides can lead to sublethal effects. Sublethal effects mean that an organism survives exposure to a toxicant at lethal or sublethal concentration. A sublethal concentration is generally considered to be less than the LC_50_ (50% Lethal Concentration) [16]. The sublethal effects may be manifested as reductions in longevity, development rates, population growth, fertility, fecundity, changes in sex ratio, deformities, changes in behavior, feeding, and oviposition [17].

Stylet behavior of sucking-type insects such as Hemiptera is difficult to observe with the naked eye, so it is detected by using an electrical device [18,19]. An electrical penetration graph (EPG) is an effective tool to study the feeding behaviors of sucking-type insects [20]. Many studies have examined the feeding behavior of sucking-type insects such as aphids, planthoppers, and whiteflies using EPGs [21,22,23,24,25]. Many studies have also been performed on the feeding behavior of BPHs. Velusamy and Heinrichs [26] reported the difference between resistant and susceptible insects using EPG for the first time. After that, other researchers studied the differences in feeding behavior among different varieties [27,28,29]. However, the sublethal effects of insecticides on the BPH and WBPH have not been addressed. In this study, we investigated the sublethal effects of imidacloprid and sulfoxaflor on the fecundity, survival, development, and feeding behavior of the BPH and WBPH. Our results provide relevant information for the optimal use of imidacloprid and sulfoxaflor against these pests.

## 2. Materials and Methods

### 2.1. Insects

The BPHs used in this study were collected in Goseong (Gyeongsangnam-do, Republic of Korea, in 2020), and the WBPHs were collected in Shinan (Jeollanam-do, Republic of Korea, in 2020). They were then reared in a laboratory on the susceptible rice variety, Chucheongbyeo, without contact with any insecticides. The conditions included a constant temperature of 25 ± 2 °C, 50–60% relative humidity (RH) and 16 L:8 D photoperiod.

### 2.2. Insecticides

Commercially formulated imidacloprid (8%, SC), sulfoxaflor (7%, SC), dinotefuran (50%, SG), carbosulfan (20%, SC), clothianidin (8%, SG), etofenprox (10%, EW), fenobucarb (50%, EC), flonicamid (10%, WG), pymetrozine (49%, WG), and thiamethoxam (10%, WG) were purchased from a farm supply store (Seowon, Cheongju, Republic of Korea). A stock solution and serial dilutions were prepared by dissolution in distilled water.

### 2.3. Bioassays

The spray method was adopted to test the toxicity of insecticides to BPH and WBPH female adults. Unmated adult females (1–2 days old) were used as test insects in this study. Ten different insecticides were diluted in distilled water to the recommended concentrations. Imidacloprid (32, 16, 8, 1.6, 0.16, and 0.016 ppm) and sulfoxaflor (140, 14, 7, 0.7, 0.07, and 0.007 ppm) were diluted to a series of concentrations in distilled water. Inoculated leaves (rice seedlings) were detached, placed in a glass test tube (30 mm internal diameter, 15 cm long). Twenty insects were placed at each concentration and insecticides were administered 5 times (500 μL) by using sprayer. The control treatments included distilled water only. The results were checked after 96 h. The LC_10_ and LC_30_ values were determined based on standard probit analysis using SAS [30]. All the experiments are repeated three times independently.

### 2.4. Sublethal Effects of Two Insecticides on Reproduction

The LC_10_ and LC_30_ concentrations of imidacloprid and sulfoxaflor were used in the biological parameter experiment. By using sprayer, insecticides were administered 10 times (1 mL) in breeding circle cage (10 × 20 cm). One unmated male adult and one unmated female adult was paired and placed separately into a breeding circle cage containing fresh rice seedlings. Ten pairs were treated, and three replicates were used in the experiment. Cages were changed every 24 h and the number of eggs was recorded by cutting the rice stem. After that, the developmental period from 1st to 5th instar nymph and the lifespan of adults were investigated. The BPH and WBPH were observed every 24 h and recorded until death. By observing the insects growing in this cage, the net reproductive rate (*R*_0_), the intrinsic rate of increase (*r_m_*), the mean generation time (*T*), the finite rate of increase (*λ*), and doubling time (*DT*) were determined [31].

### 2.5. Electrical Penetration Graph (EPG)

Feeding behaviors of BPHs and WBPHs at the sublethal concentrations were monitored by using an 8-channel direct current-electrical penetration graph (DC-EPG) [19,32,33]. Starved planthoppers were connected individually to thin gold wire (3–5 cm length, 18 μm diameter; Goodfellow, Cambridge, UK) using silver conductive paint (RS Components Ltd., Northants, UK) and connected to the EPG input probe. The voltage was supplied to each plant via a copper electrode inserted into the compost. EPG recording was carried out using a Giga-8 EPG amplifier system with 1 GΩ input resistance (EPG Systems, Wageningen, The Netherlands). A plant electrode was inserted into the cotton of each potted plant and connected to the plant voltage output of the Giga-8 EPG device. After wiring and attachment to the system, planthoppers were suspended and starved for 2 h before monitoring. Recordings were made simultaneously on eight plant placed within a Faraday cage at 25 ± 2 °C under electric fluorescent lighting. Eight insects were used per concentration and the experiment was repeated 10 times. Insects and plants were used only once and then discarded. The data were statistically analyzed using one-way ANOVA followed by Tukey’s honestly significant difference (HSD) test.

## 3. Results

### 3.1. Toxicity of 10 Insecticides to the BPH and WBPH

The mortality of the BPH and WBPH at the recommended concentrations of 10 insecticides is shown in Table 1. The lethal effects on BPHs were the highest in the fenobucarb- and dinotefuran-treated groups. These two insecticides also induced 100% mortality in the WBPH. In addition, sulfoxaflor and clothianidin induced 100% mortality in the WBPH. In both species, flonicamid resulted in the lowest mortality (29.3% for the BPH and 44.8% for the WBPH). The insecticide with the largest differences in mortality rates between the BPH and WBPH was imidacloprid. The insecticide resistance action committee (IRAC) mode of action group number of imidacloprid is four. Therefore, the same group, sulfoxaflor, was selected, and the sublethal effect was determined.

### 3.2. Selection of Sublethal Concentrations

The sublethal concentrations (LC_10_ and LC_30_) of imidacloprid and sulfoxaflor were determined (Table 2). For the BPH, the sublethal concentrations of imidacloprid were 4.04 (LC_10_) and 12.72 ppm (LC_30_), and the sublethal concentrations of sulfoxaflor were 0.83 (LC_10_) and 2.30 ppm (LC_30_). For the WBPH, the sublethal concentrations of imidacloprid were 1.32 (LC_10_) and 3.55 ppm (LC_30_), and the sublethal concentrations of sulfoxaflor were 0.43 (LC_10_) and 1.77 ppm (LC_30_).

### 3.3. Analysis of Feeding Behaviors

The changes in the feeding behavior of the BPH and WBPH were analyzed as waveforms using EPG (Figure 1). NP (nonpenetration) is a nonprobing waveform with the stylet and the host surface completely separated from each other. The N4-a phase is intracellular activity in the phloem region, which is necessary just prior to ingesting phloem sap. The N4-b phase is phloem sap ingestion, and the N5 phase is the waveform in which stylet activity occurs in the xylem region. For BPHs, the total durations of NP waveform by imidacloprid (LC_10_ and LC_30_) were 164.74 and 176.48 min, respectively, which were longer than those in control (52.73 min) (Figure 2A). The total durations of sulfoxaflor (LC_10_ and LC_30_) were 235.57 and 226.93 min, respectively, which were also longer than those of the control group. Including the control group, the total durations of the N4-a waveform for imidacloprid and sulfoxaflor were recorded within 10 min, except for imidacloprid LC_10_ (Figure 2B). The total durations of N4-b for imidacloprid (LC_10_ = 48.72, LC_30_ = 7.80 min) and sulfoxaflor (LC_10_ = 13.36, LC_30_ = 9.86 min) decreased with increasing pesticide concentrations (Figure 2C). There was no significance in the N5 waveform (Figure 2D).

For WBPHs, the total durations of NP waveform by imidacloprid (LC_10_ = 203.69 and LC_30_ = 169.9 min) and sulfoxaflor (LC_10_ = 134.02 and LC_30_ = 252.14 min) were significantly different from those in the control group (45.18 min) (Figure 3A). Except for the LC_10_ of sulfoxaflor, the total durations of the N4-a waveform were significantly different from those of the control group at other concentrations (Figure 3B). In the phloem feeding waveform N4-b (Figure 3C), the two pesticides were significantly different from the control group, but they were not in the xylem feeding waveform N5 (Figure 3D). The feeding time in the two species was significantly different in the N4-a and N4-b waveforms of imidacloprid LC_10_ (Figure 4). In the case of sulfoxaflor LC_10_, the NP waveform showed a significant difference in the BPH and WBPH, and there was no significant difference for N4-a and N4-b (Figure 5).

Table 3 shows the relationship between two insecticides and the number of occurrences of each EPG waveform. The occurrences of NP wave form significantly increased in only sulfoxaflor LC_10_ on WBPHs. The occurrences of N5 wave form significantly increased in only imidacloprid LC_10_ on WBPHs.

### 3.4. Influence of Imidacloprid and Sulfoxaflor at Sublethal Concentrations on Lifespan and Fecundity

The biological parameters following treatment of the planthoppers with sublethal concentrations of imidacloprid and sulfoxaflor are shown in Table 4. Compared with the control (20.43 days), the developmental durations of BPHs from 1st to 5th instar nymphs following treatment with the LC_30_ of sulfoxaflor were significantly decreased by 15.93 days. The adult longevity following LC_10_ and LC_30_ imidacloprid treatments was 9.0 and 6.2 days, respectively. In sulfoxaflor, it was 9.52 (LC_10_) and 6.96 days (LC_30_), which showed a significant decrease compared with the control (15.29 days). Accordingly, the total lifespan from hatching to death also decreased, and there was a significant difference between the control group (35.71 days) and the treated groups (LC_10_, 28.04 and LC_30_, 22.89 days). The lifespan of the sulfoxaflor LC_30_-treated group (22.89 days) was shorter than that of the imidacloprid LC_30_-treated group (25.64 days). The duration of pre-oviposition in BPHs was 5–7 days, which was significantly longer than the control group at the sublethal concentration of the two insecticides. The number of eggs per female adult was 129.07 in the control group but 72.27 under the imidacloprid LC_10_ and 49.93 under LC_30_, which was a significant decrease. In sulfoxaflor, the LC_10_ (66.0) and LC_30_ (36.67) were significantly different from those in the control group.

For WBPHs, there were no significant differences observed between the treatment and control groups for developmental time. However, adult longevity was decreased in the treated groups. The control group was 13.57 days, the imidacloprid LC_10_ and LC_30_ were 9.18 and 6.67, respectively, and the sulfoxaflor LC_10_ and LC_30_ were 8.96 and 6.14, respectively. There was a significant difference between LC_10_ and LC_30_. Like BPHs, the total lifespan of WBPHs from hatching to death also decreased in insecticide-treated groups. The duration of pre-oviposition was 5–7 days (average 6 days). The total fecundity was 102.33 in the control group of WBPHs, whereas the imidacloprid LC_10_ (39.93) and LC_30_ (28.13) and sulfoxaflor LC_10_ (56.07) and LC_30_ (49.07) were significantly different from the control group and further decreased under imidacloprid. Figure 6 shows the daily survival rate of the BPH and WBPH. For both imidacloprid and sulfoxaflor, the survival rate of the BPH and WBPH decreased more at the LC_30_ concentration than at the LC_10_ concentration.

### 3.5. Life Table

The net productive rate (*R*_0_), the intrinsic rate of increase (*r_m_*), the mean generation time (T), the finite rate of increase (λ), and doubling time (*DT*) were calculated and analyzed (Table 5). For BPHs, the *R*_0_ was significantly higher in the control group. The *R*_0_ of the imidacloprid LC_10_ and LC_30_ was 56.98 and 38.84, respectively. The *R*_0_ of the sulfoxaflor LC_10_ and LC_30_ was 46.20 and 21.67, respectively. The *r_m_* and *λ* showed a significant difference in all treatment groups compared to the untreated group. Except for the imidacloprid LC_10_, *T* decreased with pesticide treatment compared to the control group. The *DT* was highest at imidacloprid LC_30_ (5.81). For WBPHs, the *R*_0_ was decreased in the pesticide-treated group compared to the control group (100.83), as was observed for BPHs. In particular, the *R*_0_ of the imidacloprid-treated groups (LC_10_ = 31.67 and LC_30_ = 21.25) decreased more than that of the sulfoxaflor-treated groups (LC_10_ = 42.15 and LC_30_ = 30.66). The *T* was lower in the pesticide-treated groups (imidacloprid LC_10_ = 22.91, LC_30_ = 21.06 and sulfoxaflor LC_10_ = 22.12, LC_30_ = 20.29) than in the control group (24.24), and its value was inversely proportional to the concentration. *λ* also decreased with pesticide treatment. The *DT* was lowest in the control group (3.65) and highest in the imidacloprid LC_30_ group (4.82).

## 4. Discussion

Commonly, sublethal concentrations reduce the feeding, development, survival, and fertility of insects [34,35,36,37,38,39]. In this study, treatment with imidacloprid and sulfoxaflor resulted in certain changes in the physiological and biological characteristics of BPHs and WBPHs. EPG was used to analyze the feeding behaviors of the BPH and WBPH at sublethal concentrations. There was no difference between the control group and the treated group in the number of NP waveforms (Figure 2A and Figure 3A). However, the waveforms before phloem feeding (N4-a) and during phloem feeding (N4-b) were generally significantly decreased when the insecticides were applied (Figure 2B,C and Figure 3B,C). In addition, as the concentration increased, the number of recordings of phloem feeding-related waveforms decreased (Figure 4 and Figure 5). Before the insect stylet reaches the phloem, multiple attempts are made to find a suitable site for feeding. Previously, resistant cotton aphids (*Aphis gossypii)* were found to be more active in finding suitable feeding sites on imidacloprid-treated hosts than susceptible cotton aphids [40]. In our experiment, the BPH was more resistant to imidacloprid and more active in phloem feeding, but there was no significant difference with sulfoxaflor. However, the attempts to feed on the phloem by the BPH and WBPH as affected by the sublethal effect of imidacloprid and sulfoxaflor decreased; accordingly, the phloem feeding time was also reduced. In a previous study, it was reported that treatment with imidacloprid reduced phloem feeding by *Bactericera cockerelli* on the host plant [41] and that treatment with sulfoxaflor reduced *Myzus persicae* phloem feeding [42]. For the BPH and WBPH, as in the previous study, when imidacloprid and sulfoxaflor were treated at sublethal concentrations, the phloem feeding time was reduced. It was found that imidacloprid had a better effect on the inhibition of feeding of the WBPH, and sulfoxaflor showed a better effect on the inhibition of feeding of the BPH. Therefore, it is suggested that treatment with sublethal concentrations of the imidacloprid and sulfoxaflor will reduce the feeding of BPHs and WBPHs on rice phloem. This reduction in feeding will consequently affect the survival rate and reproduction of the insects.

In fact, according to the results of this study, the fecundity and lifespan of the BPH and WBPH decreased at sublethal concentrations (Table 4). While there was a study result showing that the developmental time of *Diaphorina citri* nymphs increased with sublethal concentrations of imidacloprid [43], there was a study showing the opposite result in *Myzus persicae* [34]. This suggests that even the same insecticide may have different effects on different insect species. In this study, the developmental period of nymphs, adult longevity, and fecundity were all decreased in the insecticide-treated group; accordingly, the *R*_0_ and *T* also decreased. A study reported that the fertility of the BPH decreased at a sublethal concentration of paichongding, a neonicotinoid insecticide [44]. In addition, a study showed that the reproduction of BPHs was reduced by a sublethal concentration of sulfoxaflor [45]. Likewise, in this study, the *R*_0_, *r_m_*, and *T* of BPHs and WBPHs were significantly decreased in the imidacloprid- and sulfoxaflor-treated group compared to the control group. In particular, the *R*_0_ was reduced by more than half in the treated group compared to the control group. In both the BPH and WBPH, the *T* decreased more in the group treated with sulfoxaflor than in the group treated with imidacloprid because the lifespan of the BPH and WBPH was further reduced when treated with sulfoxaflor. Additionally, the *DT* was different due to fecundity.

## 5. Conclusions

The sublethal concentrations of imidacloprid and sulfoxaflor inhibited the phloem feeding of BPHs and WBPHs and decreased their *R*_0_, *r_m_*, and *T*. Additionally, sulfoxaflor was effective in inhibiting feeding in the BPH, and imidacloprid was more effective in inhibiting feeding in the WBPH. However, it should be noted that this result does not indicate a difference between the species. In other words, it should not be concluded that imidacloprid is more effective against the WBPH and sulfoxaflor is more effective against the BPH. Additional studies are needed to determine whether the effects of the pesticides differ between species or strains. Pesticide treatment at sublethal concentrations affects the reproduction of insects, and low doses are relatively safe for natural enemies and reduce damage to the environment and livestock [46]. The data from this study will be utilized as basic data to facilitate more rational use of imidacloprid and sulfoxaflor for BPH and WBPH control. However, all experiments were carried out in the laboratory and further field research needs to be conducted.

## Figures and Tables

**Figure 1 insects-13-01130-f001:**
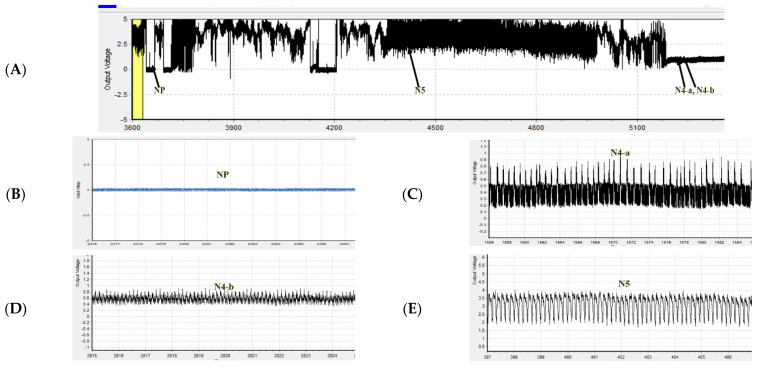
EPG waveforms of *Nilaparvata lugens* and *S. furcifera* adults. (**A**) Overall typical view for one hour and a half; (**B**) NP: nonpenetration; (**C**) N4-a: intracellular activity in the phloem region, which is necessary just prior to the ingestion of phloem sap; (**D**) N4-b: phloem sap ingestion; (**E**) N5: stylet activity in the xylem region.

**Figure 2 insects-13-01130-f002:**
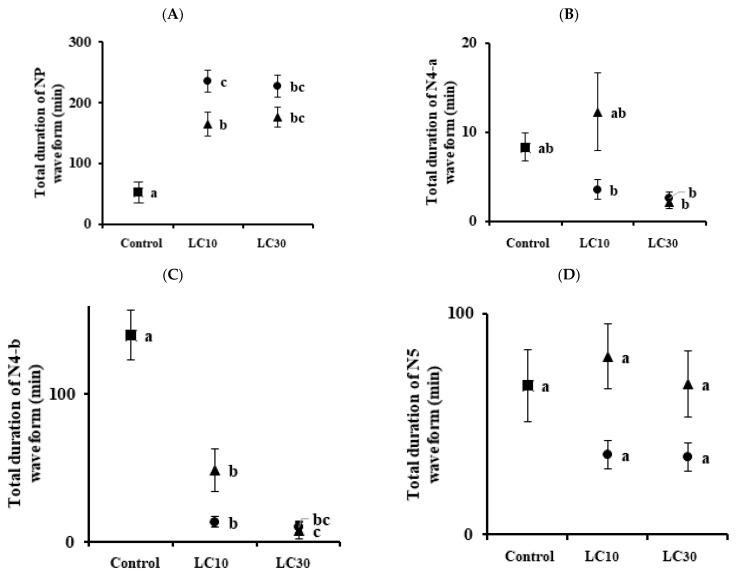
Relationship between two insecticide concentrations and the total duration of each EPG waveform for *N. lugens*. Panel (**A**) is the duration of the NP waveform, panel (**B**) is the N4-a waveform, panel (**C**) is the N4-b waveform, and panel (**D**) is the N5 waveform. Triangles are imidacloprid-treated groups, and circles are sulfoxaflor-treated groups. Means followed by the same letters are not significantly different (*p* = 0.05; Tukey’s studentized range test (SAS Institute 9.0 (SAS, 2009)). Error bars indicate ± SE.

**Figure 3 insects-13-01130-f003:**
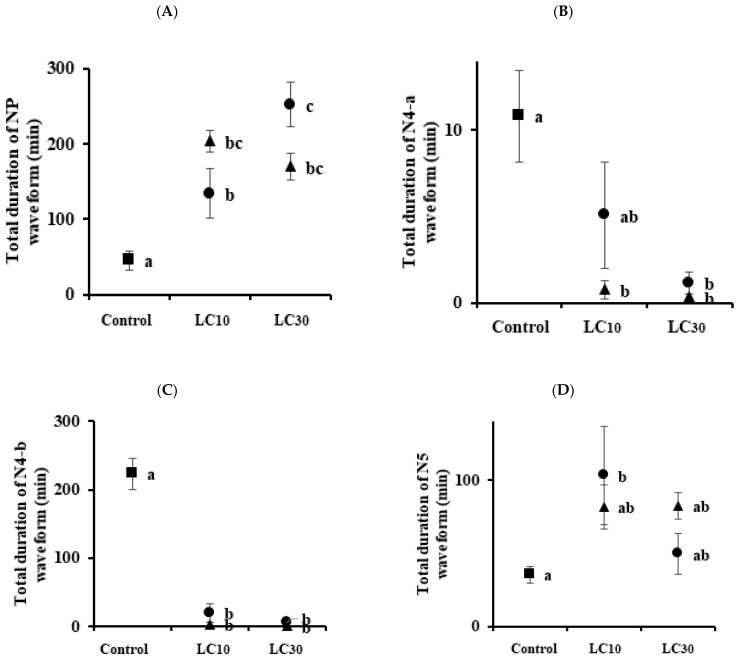
Relationship between two insecticide concentrations and the total duration of each EPG waveform for *S. furcifera*. Panel (**A**) is the duration of the NP waveform, panel (**B**) is the N4-a waveform, panel (**C**) is the N4-b waveform, and panel (**D**) is the N5 waveform. Triangles are imidacloprid-treated groups and circles are sulfoxaflor-treated groups. Means followed by the same letters are not significantly different (*p* = 0.05; Tukey’s studentized range test (SAS Institute 9.0 (SAS, 2009)). Error bars indicate ± SE.

**Figure 4 insects-13-01130-f004:**
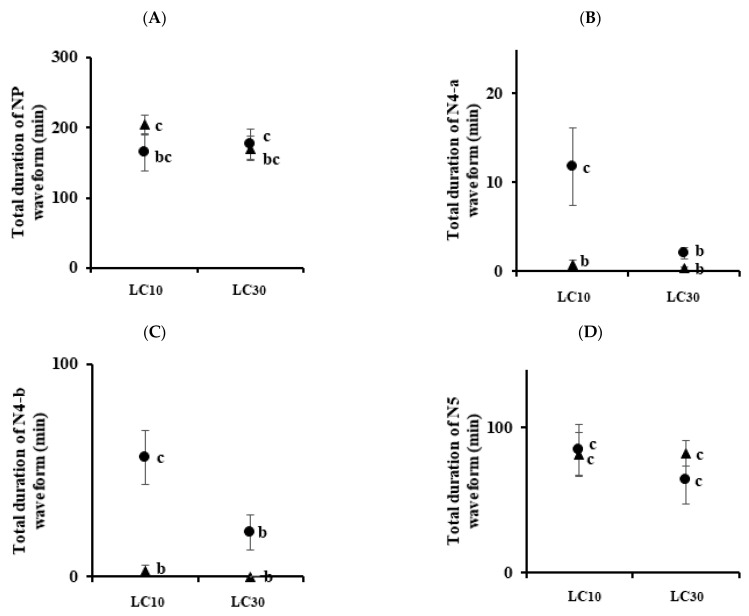
Total duration for each waveform of *N. lugens* and *S. furcifera* treated with LC_10_ and LC_30_ concentrations of imidacloprid. Panel (**A**) is the duration of the NP waveform, panel (**B**) is the N4-a waveform, panel (**C**) is the N4-b waveform, and panel (**D**) is the N5 waveform. Circles are *Nilaparvata lugens* and triangles are *Sogatella furcifera*. Means followed by the same letters are not significantly different (*p* = 0.05; Tukey’s studentized range test (SAS Institute 9.0 (SAS, 2009)). Error bars indicate ± SE.

**Figure 5 insects-13-01130-f005:**
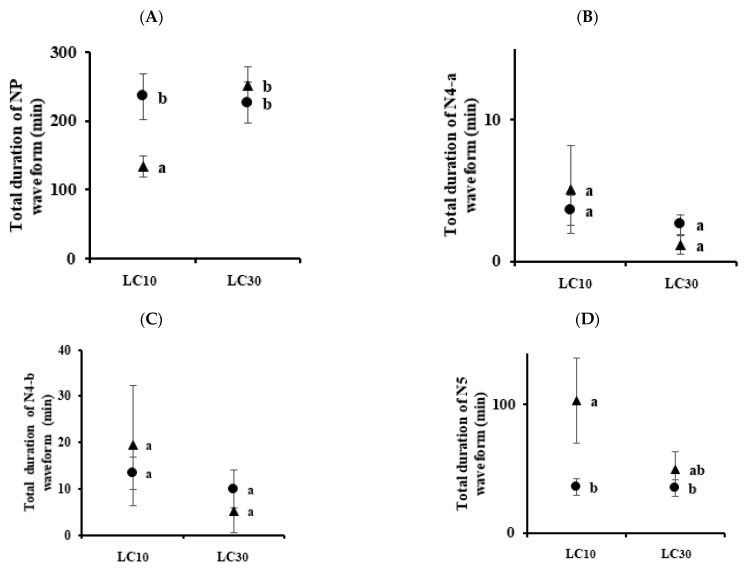
Total duration for each waveform of *N. lugens* and *S. furcifera* treated with LC_10_ and LC_30_ concentrations of sulfoxaflor. Panel (**A**) is the duration of the NP waveform, panel (**B**) is the N4-a waveform, panel (**C**) is the N4-b waveform, and panel (**D**) is the N5 waveform. Circles are *Nilaparvata lugens* and triangles are *Sogatella furcifera*. Means followed by the same letters are not significantly different (*p* = 0.05; Tukey’s studentized range test (SAS Institute 9.0 (SAS, 2009)). Error bars indicate ± SE.

**Figure 6 insects-13-01130-f006:**
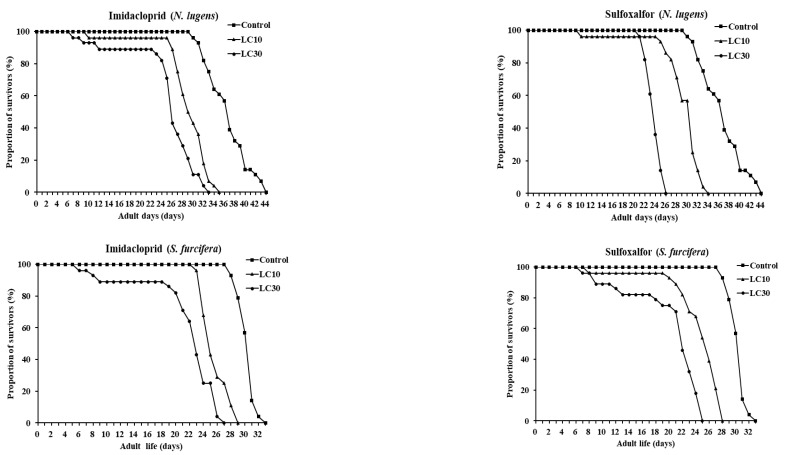
Survival rates (%) of *N. lugens* (**top**) and *S. furcifera* (**bottom**) affected by sublethal concentrations of the two insecticides. Squares are untreated-, triangles are LC_10_- and circles are LC_30_-treated group. The vertical axis represents daily survival rates of *N. lugens* and *S. furcifera* from the first-instar nymph to the death of the adults.

**Table 1 insects-13-01130-t001:** Comparison of susceptibility of *Nilaparvata lugens* and *Sogatella furcifera* at the recommended concentrations of 10 insecticides.

Insect	Insecticides	RC (ppm)	*n* ^(a)^	Mortality (%)			
				24 h	48 h	72 h	96 h
*Nilaparvata lugens*	Imidacloprid (8%, SC)	16	58	28.1 ± 14.1	39.8 ± 12.9	50.6 ± 19.5	57.5 ± 8.4
	Sulfoxaflor (7%, SC)	14	59	62.2 ± 3.8	85.8 ± 5.5	93.1 ± 11.5	93.1 ± 11.5
	Dinotefuran (50%, SG)	100	58	90.0 ± 10.0	96.6 ± 5.8	100	100
	Carbosulfan (20%, SC)	200	60	47.0 ± 18.5	59.8 ± 9.6	63.6 ± 14.0	63.6 ± 14.0
	Clothianidin (8%, SG)	16	60	63.9 ± 21.6	66.1 ± 23.6	69.3 ± 26.8	76.2 ± 15.7
	Etofenprox (10%, EW)	100	59	67.8 ± 11.3	70.5 ± 5.7	70.5 ± 5.7	70.5 ± 5.7
	Fenobucarb (50%, EC)	500	62	100	100	100	100
	Flonicamid (10%, WG)	20	63	10.0	17.2 ± 17.3	20.7 ± 20.8	29.3 ± 7.6
	Pymetrozine (49%, WG)	98	61	32.1 ± 10.7	56.1 ± 16.5	66.5 ± 15.6	69.9 ± 10.1
	Thiamethoxam (10% WG)	10	60	50.0 ± 10.0	48.3 ± 10.0	55.2 ± 15.3	58.6 ± 10.0
*Sogatella furcifera*	Imidacloprid (8%, SC)	16	60	56.7 ± 5.8	66.7 ± 5.8	80.0	96.6 ± 5.8
	Sulfoxaflor (7%, SC)	14	60	40.0 ± 10.0	60.0 ± 10.0	83.3 ± 5.8	100
	Dinotefuran (50%, SG)	100	60	93.3 ± 11.5	100	100	100
	Carbosulfan (20%, SC)	200	60	90.0 ± 10.0	90.0 ± 10.0	90.0 ± 10.0	89.7 ± 10.0
	Clothianidin (8%, SG)	16	60	86.7 ± 5.8	93.3 ± 5.8	100	100
	Etofenprox (10%, EW)	100	60	70.0 ± 10.0	70.0 ± 10.0	70.0 ± 10.0	89.7 ± 10.0
	Fenobucarb (50%, EC)	500	60	100	100	100	100
	Flonicamid (10%, WG)	20	60	6.7 ± 11.5	13.3 ± 5.8	30.0 ± 10.0	44.8 ± 20.8
	Pymetrozine (49%, WG)	98	60	13.3 ± 5.8	36.7 ± 11.5	50.0 ± 17.3	65.5 ± 15.3
	Thiamethoxam (10% WG)	10	60	30.0 ± 17.3	56.7 ± 15.3	70.0 ± 30.0	96.6 ± 5.8

^(a)^ *n* is the total number of tested insects.

**Table 2 insects-13-01130-t002:** Toxicity of two insecticides against *Nilaparvata lugens* and *Sogatella furcifera* adults.

Insect	Insecticides	*n* ^(^ ^a)^	Mortality (%)	LC ^(b)^_10_	LC_30_	Slope ± SE ^(c)^
*Nilaparvata lugens*	Imidacloprid	244	39.8	4.04 (2.73–5.47)	12.72 (10.01–15.64)	0.18 ± 0.12
	Sulfoxaflor	240	85.8	0.83 (0.60–1.08)	2.30 (1.86–2.75)	0.10 ± 0.12
*Sogatella furcifera*	Imidacloprid	207	66.7	1.32 (0.89–1.80)	3.55 (2.75–4.37)	0.15 ± 0.14
	Sulfoxaflor	167	60.0	0.43 (0.14–0.87)	1.77 (0.87–2.91)	0.14 ± 0.13

^(a)^ *n* is the total number of tested insects. ^(b)^ LC is the lethal concentration. ^(c)^ SE is the standard error.

**Table 3 insects-13-01130-t003:** Relationship between the two insecticide concentrations and the number of occurrences of each EPG waveform for *Nilaparvata lugens* and *Sogatella furcifera*.

Insect	Insecticides	Conc.	NP	N4-a	N4-b	N5
*Nilaparvata lugens*	Control		9.88 ± 0.97 a ^(a)^	6.25 ± 1.0 a	5.25 ± 0.88 a	2.75 ± 0.53 a
	Imidacloprid	LC_10_	11.13 ± 1.01 a	3.0 ± 0.93 ab	2.38 ± 0.73 b	5.38 ± 0.84 a
		LC_30_	9.50 ± 1.16 a	0.88 ± 0.40 b	0.88 ± 0.40 b	5.55 ± 0.80 a
	Sulfoxaflor	LC_10_	10.0 ± 1.46 a	3.0 ± 0.89 ab	1.38 ± 0.32 b	4.88 ± 0.95 a
		LC_30_	9.38 ± 1.16 a	2.38 ± 0.68 b	0.88 ± 0.35 b	4.75 ± 0.82 a
*Sogatella furcifera*	Control		11.63 ± 1.97 a	4.88 ± 1.06 a	2.50 ± 0.57 a	4.50 ± 0.68 a
	Imidacloprid	LC_10_	14.13 ± 1.42 ab	1.38 ± 0.32 b	0.63 ± 0.26 b	9.50 ± 1.52 b
		LC_30_	11.25 ± 1.10 a	0.75 ± 0.31 b	0.0 b	7.50 ± 1.24 ab
	Sulfoxaflor	LC_10_	18.63 ± 1.31 b	5.0 ± 1.02 a	1.25 ± 0.25 ab	8.38 ± 1.0 ab
		LC_30_	10.88 ± 2.21 a	2.25 ± 0.62 ab	0.50 ± 0.19 b	4.38 ± 1.03 a

^(a)^ Means followed by the same letters are not significantly different (*p* = 0.05; Tukey’s studentized range test (SAS Institute 9.0 (SAS, 2009)).

**Table 4 insects-13-01130-t004:** Developmental duration, longevity, and fecundity of *Nilaparvata lugens* and *Sogatella furcifera* on rice.

Stage	*Nilaparvata lugens*					*Sogatella furcifera*				
	Control	Imidacloprid		Sulfoxaflor		Control	Imidacloprid		Sulfoxaflor	
		LC_10_	LC_30_	LC_10_	LC_30_		LC_10_	LC_30_	LC_10_	LC_30_
1st to 5th instar (days)	20.43 ± 0.39 a ^(a)^	19.57 ± 0.60 ab	18.0 ± 0.91 bc	18.86 ± 0.49 ab	15.93 ± 0.10 c	15.82 ± 0.09 a	15.54 ± 0.11 a	14.5 ± 0.61 a	15.04 ± 0.45 a	14.39 ± 0.59 a
Adult longevity (days)	15.29 ± 0.74 a	9.0 ± 0.20 b	6.2 ± 0.41 c	9.52 ± 0.26 b	6.96 ± 0.26 c	13.57 ± 0.19 a	9.18 ± 0.31 b	6.67 ± 0.34 c	8.96 ± 0.34 b	6.14 ± 0.34 c
Total longevity (days)	35.71 ± 0.74 a	28.61 ± 0.60 b	25.64 ± 0.60 c	28.04 ± 0.59 b	22.89 ± 0.26 d	29.39 ± 0.26 a	24.71 ± 0.34 b	20.21 ± 1.08 cd	23.36 ± 0.93 bc	19.21 ± 1.09 d
Pre-oviposition (days)	5.5 ± 0.14 a	7.64 ± 0.09 c	7.54 ± 0.10 c	7.32 ± 0.09 c	5.96 ± 0.11 b	5.82 ± 0.07 a	5.64 ± 0.09 ab	6.04 ± 0.13 ab	6.14 ± 0.07 ab	6.18 ± 0.07 ab
Fecundity (eggs/female)	129.07 ± 9.83 a	72.27 ± 3.83 b	47.93 ± 3.58 cd	66.0 ± 3.54 bc	36.67 ± 3.09 d	102.33 ± 6.11 a	39.93 ± 2.83 cd	28.13 ± 1.96 d	56.07 ± 2.81 b	49.07 ± 2.61 bc

^(a)^ Means followed by the same letters are not significantly different (*p* = 0.05; Tukey’s studentized range test (SAS Institute 9.0 (SAS, 2009)).

**Table 5 insects-13-01130-t005:** The mean rates of stable population parameters of *Nilaparvata lugens* and *Sogatella furcifera*.

Insect	Insecticides		*R*_0_ ^(a)^	*r_m_* ^(b)^	*T* ^(c)^	*λ* ^(d)^	*DT* ^(e)^
*Nilaparvata lugens*	Control		148.05 ± 10.19 a ^(f)^	0.182 ± 0.004 a	27.23 ± 0.30 a	1.202 ± 0.004 a	3.80 ± 0.08 a
	Imidacloprid	LC_10_	56.98 ± 3.07 b	0.140 ± 0.007 b	27.34 ± 0.56 ab	1.149 ± 0.008 bc	5.09 ± 0.26 c
		LC_30_	38.84 ± 1.60 bc	0.121 ± 0.005 c	25.23 ± 0.25 c	1.129 ± 0.005 c	5.81 ± 0.20 d
	Sulfoxaflor	LC_10_	46.20 ± 4.96 b	0.157 ± 0.003 b	25.94 ± 0.17 bc	1.167 ± 0.003 b	4.48 ± 0.09 bc
		LC_30_	21.67 ± 2.20 c	0.148 ± 0.003 b	20.74 ± 0.20 d	1.162 ± 0.004 b	4.12 ± 0.07 b
*Sogatella furcifera*	Control		100.83 ± 5.27 a	0.19 ± 0.003 a	24.24 ± 0.18 a	1.209 ± 0.003 a	3.65 ± 0.05 a
	Imidacloprid	LC_10_	31.67 ± 2.13 bc	0.151 ± 0.004 c	22.91 ± 0.24 b	1.163 ± 0.004 c	4.65 ± 0.13 c
		LC_30_	21.25 ± 1.19 c	0.145 ± 0.003 c	21.06 ± 0.23 c	1.156 ± 0.005 c	4.82 ± 0.13 c
	Sulfoxaflor	LC_10_	42.15 ± 1.76 b	0.169 ± 0.003 b	22.12 ± 0.30 b	1.185 ± 0.004 b	4.12 ± 0.08 b
		LC_30_	30.66 ± 2.00 c	0.169 ± 0.005 b	20.29 ± 0.19 c	1.183 ± 0.005 b	4.15 ± 0.10 b

^(a)^*R*_0_, Net reproductive rate; ^(b)^ *r_m_*, the intrinsic rate of increase; ^(c)^ *T*, the mean generation time; ^(d)^ *λ*, the finite rate of increase; ^(e)^ *DT*, Doubling time. ^(f)^ Means followed by the same letters are not significantly different (*p* = 0.05; Tukey’s studentized range test (SAS Institute 9.0 (SAS, 2009)).

## Data Availability

The data presented in this study are available on request from the corresponding author.

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
