# Peer review of "Analysis of the Feeding Behavior and Life Table of Nilaparvata lugens and Sogatella furcifera (Hemiptera: Delphacidae) under Sublethal Concentrations of Imidacloprid and Sulfoxaflor"

_insects, 2022, doi:10.3390/insects13121130_

Round 1

Reviewer 1 Report

Manuscript " Analysis of the Feeding Behavior and Life Table of Nilapar-vata lugens and Sogatella furcifera (Hemiptera: Delphacidae) under Sublethal Concentrations of Imidacloprid and Sul-foxaflor " was used to study the change of feeding behavior of brown planthopper and white backed planthopper after treatment with imidacloprid and sulfoxaflor by EPG technology. The results can provide new evidence for understanding the effects of insecticides on insect feeding behavior. However, the sublethal effects of imidacloprid and sulfoxaflor on brown planthopper and white backed planthopper have been reported (Bao et al., 2009; Zhou et al., 2017; Xiang et al., 2019; Liao et al., 2019). What is the significance of the author's research again? In addition, a large number of studies have shown that brown planthopper and white backed planthopper have high resistance to imidacloprid. Why did the author choose imidacloprid for research? Moreover, in a large number of previous studies, the number of laid eggs by females of brown planthopper and white backed planthopper in the control group was significantly higher than that in this study. What is the reason that brown planthopper and white backed planthopper laid so few eggs in the control group? The specific opinions are as follows:

Zhou C, Liu L, Yang H, et al. Sublethal effects of imidacloprid on the development, reproduction, and susceptibility of the white-backed planthopper, Sogatella furcifera (Hemiptera: Delphacidae). Journal of Asia-pacific entomology, 2017, 20(3): 996-1000.

Bao H, Liu S, Gu J, et al. Sublethal effects of four insecticides on the reproduction and wing formation of brown planthopper, Nilaparvata lugens. Pest Management Science: Formerly Pesticide Science, 2009, 65(2): 170-174.

Xiang X, Liu S, Wang X, et al. Sublethal effects of sulfoxaflor on population projection and development of the white-backed planthopper, Sogatella furcifera (Hemiptera: Delphacidae). Crop Protection, 2019, 120: 97-102.

Liao X, Ali E, Li W, et al. Sublethal effects of sulfoxaflor on the development and reproduction of the brown planthopper, Nilaparvata lugens (Stål). Crop Protection, 2019, 118: 6-14.

Line 20-23. These results are not very concerned by readers. In this manuscript, readers may prefer to see more descriptions of EPG results.

Line 72. Why is it that commercially formulated is used for research, rather than the technical insecticide used?

Line 79. A brief introduction to the "array method" is required.

Line 83. filled with water”?

Line 88-96. The method description is too simple. For example, which method is used for treatment. How to count the number of laid eggs by females.

Line 120. Abbreviations need to be defined when they first appear. The full name of "IRAC" is required. In addition, where can we clearly know that "the IRAC mode of action group number of imidacloprid is 4" ?

Line 119-120. Why should the insecticide with the largest difference in mortality between BPH and WBPH be selected for sublethal effect determination?

Table 1. What do GS and RDA represent respectively?

Line 145-146. “shorter than”? Please check carefully. Change "Figure 2" to "Figure 2A".

Figure 2. Legend needs to be added.

Table 4. Why did the control group laid so few eggs? Xu et al. (2019) reported that the number of laid eggs by female of brown planthopper was 471.0627. Xiang et al. (2019) reported that the number of laid eggs by female of white-backed planthopper was 168.1.

Xu P, Shu R, Gong P, et al. Sublethal and transgenerational effects of triflumezopyrim on the biological traits of the brown planthopper, Nilaparvata lugens (Stål)(Hemiptera: Delphacidae). Crop Protection, 2019, 117: 63-68.

Discussion. Need to rewrite. According to different experimental results, for example, EPG, development, reproduction and other aspects are discussed respectively.

Author Response

We appreciate for the consideration of three anonymous reviewer and final correction. All comments of reviewer on the manuscript were accepted and all comments are changed properly.

Thank you so much for your kind comments.

Reviewer 2 Report

This manuscript investigated the effects of sublethal concentrations of imidacloprid and sulfoxaflor on the feeding behavior and life table parameters of Nilaparvata lugens and Sogatella furcifera. The overall structure of the article is complete and the amount of data is sufficient, which provides a reference for the more rational use of imidacloprid and sulfoxaflor for the prevention and treatment of BPH and WBPH. However, there are still many problems with the article, requiring major revisions before publication.

1.     Simple summary: suggest to add your research background and significance;

2.     Abstract: suggest to add your research significance in the last part;

3.     Keywords: suggest to add “imidacloprid” and “sulfoxaflor”;

4.     L39-42: description of the way in which the two planthoppers harm is not comprehensive, they can also cause harm by spreading viruses and other ways;

5.     Suggest L42-“One of the methods for controlling …” to L51-“…and longevity of an organism” as a paragraph; and put the section about EPG in another paragraph;

6.     L44-“Pesticides such as imidacloprid, sulfoxaflor, clothianidin, and dinotefuran…”, add the references of this sentence;

7.     L48-50: description of “sublethal effects” is inaccurate, please rewrite;

8.     L85: describe how to determine insect death;

9.     L110-111: please give details of the number and the instar of EPG experimental insects;

10.  L115-116: “The lethal effects on BPHs were the highest in the fenobucarb- and dinotefuran-treated groups.” This comparison is not meaningful and there is no uniform concentration for comparison;

11.  L119-120: “The insecticide with the largest differences in mortality rates between the BPH and WBPH was imidacloprid.” Is there a statistical basis?

12.  L120: IRAC? Please describe more;

13.  Table 1: “na”- add the notes;

14.  Section 3.2 and Table 2: Why choose 48 hours for mortality statistics, not clear how to calculate LC10 and LC30;

15.  Table 3: This table should not be placed in this position and is not described in the text;

16.  L145, L147: “shorter”? or longer, please check;

17.  Figure 2, 3, 4, 5: suggest use one-way analysis of variance (ANOVA) followed by Tukey’s honestly significant difference (HSD) test to analyze data;

18.  L180: “…no significant difference for N4-a, N4-b and N5.” N5 has statistically significant differences;

19.  L200: “The duration of preoviposition was between five and seven days.” Preoviposition has statistically significant differences that can be described;

20.  L202, L203: “36.67” and “47.93”- the number of eggs laid after treatment with LC30 concentration of the two pesticides is not consistent with the number on the table 4, please verify the correction;

21.  L224-225: “The DT was highest at imidacloprid LC30 (5.81) and lowest at sulfoxaflor LC10 (4.12)”, The DT was lowest in the control group (3.80);

22.  L238-288: Please divide the discussion into paragraphs;

23.  L238-245: This description of imidacloprid and sulfoxaflor should appear in the Introduction, not Discussion;

24.  L255: Please add the Latin name for “cotton aphid”;

25.  L272, L274: “there is” change to “there was”;

26.  L281-288: Please discuss with related literatures;

27.  References: Most of the literature used in the introduction is too old and more references to recent developments are recommended;

28.  L342: “Acyrthosiphon pisum” should be italicized;

29.  L348: “Nephotettix virescens” should be italicized;

30.  L367-370: format of the cited references is inconsistent with other, please unify;

31.  Please unify the order in the parts of the full text. For example, the Materials and Methods section EPG experiment is behind the life table, while the Results section EPG experiment is ahead of the life table.

Author Response

(The authors gave the same response as above.)

Reviewer 3 Report

The manuscript by Kang et al provides interesting results of sublethal effects of neonicotinoids on significant insect pests of rice crops. The manuscript will be of interest to those studying actions of insecticides on whole organisms. Below are suggestions to make sure the manuscript is understandable to as wide a readership as possible:

1/ Materials and Methods - a section of how statistical analysis was carried out should be included.

2/ Line 114 - 'at the recommended concentrations of 10 insecticides', citations should be provided detailing these recommended concentrations.

3/ Table 1 - in the legend, describe what SC, SG, EW, EC and WG mean.

4/ Table 2 - Describe what the ranges are (the values in brackets).

5/ The curves from which LC10 and LC30 were determined should be included, perhaps in a supplementary file.

6/ Table 3 - The data presented here does not seem to be described in the text.

7/ Line 151 - the data for N5 should also be summarised.

8/ Line 178 - '...waveforms of imidacloprid LC10 ...' Figure 4 should be referred to here.

9/ Line 180 - Is there not significant difference at LC10 (Fig. 5D)?

10/ Figures 2, 3, 4 and 5 Legends - describe what is denoted by the small letters (presumably significant difference?).

11/ Line 193 '... significantly decreased to 15.93 days.'?

12/ Lines 194 and 195 '...was 9.0 and 6.2 days...' are these significantly different to the control?

13/ Lines 198 and 199 - are these values significantly different to the control?

14/ Line 202 '...and 36.67...', should this be 47.93?

15/ Line 203 '...(47.93)..." should this be 36.67?

16/ Section 3.4 is a rather long paragraph. A suggestion is to start a new paragraph at Line 203 where the sentence starts with 'For WBPHs...'.

17/ Line 214 - the data in Figure 6 should be summarised in the text.

18/ Table 4 - Legend, units for Developmental time, longevity should be given.

19/ Table 5 - Units, where appropriate, should be given, eg. for average generation lifespan etc.

20/ All Tables - values that are significantly different to the control should be indicated.

21/ Discussion - this is a long section that should be broken up into more paragraphs. A suggestion is to start a new paragraph at Line 270 with the sentence starting 'In fact, according to the results...'

22/ Discussion from line 250, refer to the relevant Figures or Tables when discussing findings.

23/ Line 298 - '...relatively safe of natural enemies...', citations should be provided that support this statement. Have similar studies looking at sublethal concentrations been performed on these natural enemies?

Author Response

(The authors gave the same response as above.)

Round 2

Reviewer 1 Report

After the author's revision, the quality of the manuscript has been greatly improved and basically meets the requirements of publication. However, the author still has one question not answered clearly. Why is the number of eggs laid in the control group far lower than that previously reported?

Author Response

Thank you for your comments. 

I think so about your question.

After field collection, it was reared indoors for a long time to increase the population. Therefore, it seems that the spawning population was smaller than the results of previous studies due to the weakening of the population. However, the average of the untreated group exceeded 100, so I think it is not a small number. I will make more observations on the matter in future experiments.

Thank you very much.

Reviewer 2 Report

Minor comments:

L50, SRBSDV, RGSV, RRSV, give full names.

L112-113, rate of natural increase (rm), average generation lifespan (T), geometric rate of increase (λ) >> the intrinsic rate of increase (rm), the mean generation time (T), the finite rate of increase (λ), check all and revise.

Author Response

Thank you for your comments. I have corrected everything according to your comments. 
